# Soil-feeding termites build islands of soil physical and chemical fertility in pastures in Colombian Amazon

Ervin Humprey Duran-Bautista[1,2]*, Adriana M. Silva-Olaya[3,4], María P. Llanos-Cabrera[3,4], Katherin Yalanda-Sepúlveda[2], Juan Carlos Suárez[1,3]

1 Programa Ingeniería Agroecológica, Facultad de Ingeniería, Universidad de la Amazonía, Florencia, Caquetá, Colombia, 2 Laboratorio de ecología del suelo, Grupo de Investigación Agroecosistemas y Conservación en Bosques Amazónicos-GAIA, Florencia, Caquetá, Colombia, 3 Centro de Investigaciones Amazónicas CIMAZ Macagual César Augusto Estrada Gonzáles, Grupo de Investigación Agroecosistemas y Conservación en Bosques Amazónicos-GAIA, Florencia, Caquetá, Colombia, 4 Laboratorio de procesos biogeoquímicos, Grupo de Investigación Agroecosistemas y Conservación en Bosques Amazónicos-GAIA, Florencia, Caquetá, Colombia

* e.duran@udla.edu.co

## Abstract

Soil-feeding termite species, such as *Patawatermes turricola* (Silvestri, 1901), construct mounds that significantly alter soil properties; this species is an abundant mound-builder in the Amazon region. This study evaluated physicochemical changes in termite mounds and in *Urochloa decumbens* (Stapf) R. Webster pastures with scattered trees and rotational grazing at the Centro de Investigaciones Amazónicas Macagual - César Augusto Estrada González in the northwestern Colombian Amazon. This region receives approximately 3793 mm of annual rainfall and is characterized by Ultisol soils, which are acidic, cation-poor, and clay-dominated, where the termite *P. turricola* is predominant. The study hypothesized that mound-building activities by soil-feeding termites improve the soil physical and chemical properties by creating island of soil fertility. To test this hypothesis, we collected soil samples from the outer mound wall and unmodified topsoil 5 m away to determine porosity, bulk density, aggregate stability, aggregate-size distribution, cations, organic carbon, available phosphorus, and exchangeable acidity, these physicochemical characteristics were selected due to their potential impact on soil fertility and ecosystem function. Our results showed that termites significantly improved soil aggregate stability, as reflected in a higher weighted mean diameter value in the mound (3.88 mm) than in unmodified topsoil (3.57 mm). Macro-porosity was also higher in the mound (18.49% vs. 11.47%). Higher content of soil cations, available phosphorus, and organic C was also detected in soil mound than unmodified topsoil, as well as higher soil organic carbon (27.1 g kg$^{-1}$ vs. 23.3 g kg$^{-1}$). In contrast, exchangeable acidity was higher in surrounding soil. The mound soil presented a positive impact on soil fertility and structure compared to the adjacent topsoil. The findings obtained support

**Data availability statement:** The data set related to this study has been openly published at Zenodo https://doi.org/10.5281/zenodo.13973050

**Funding:** This work was funded by Universidad de la Amazonia through project 600.6.6331. The funders had no role in study design, data collection and analysis, decision to publish, or preparation of the manuscript.

**Competing interests:** The authors have declared that no competing interests exist.

the hypothesis that the mound-building activity of *P. turricola* termites significantly improves the physical and chemical properties of the soil by creating islands of fertility in nutrient-poor agroecosystems.

## Introduction

Termites are social insects that play a crucial role in different ecosystems worldwide, representing 40–65% total biomass of soil macro-fauna in both wet and dry tropical environments [1,2]. Through constructing soil biogenic structures, termites act as ecosystem engineers, modifying the physical, chemical, and biological properties of the soil [3]. These modifications allow termites to partially control their living environment, enabling them to remain active even in harsh conditions [2,4,5]. Their activities are particularly relevant in tropical ecosystems, contributing significantly to soil formation and nutrient cycling.

A particularly interesting group is soil-feeding termites, such as *Patawatermes turricola*. Unlike other important groups in the neotropical region, such as wood-feeding or grass-feeding species, these termites feed directly on the humic component of the soil. Soil-feeding termites construct their mounds using fecal matter mixed with inorganic soil particles [3]. This unique building process results in mounds with distinctive physico-chemical properties, including higher concentrations of organic matter, cations [6], and silt content [7]. These characteristics make termite mounds markedly different from the surrounding soil.

Previous studies have shown that termite mounds exhibit physical properties (e.g., water holding capacity, bulk density, structural stability) and chemical properties (e.g., cation exchange capacity, organic matter content, and quality) that differ from those of the adjacent soil [2,8,9]. However, the magnitude of these differences depends on several factors, including the termite species, feeding habits, soil properties, mound age, vegetation, and land use [10–13]. These differences highlight the potential role of termites in improving soil fertility, particularly in ecosystems with minimal soil disturbance and low agrochemical inputs, such as pastures and savannas [14].

This role is especially relevant in the Amazon basin, where extensive livestock farming has resulted in the establishment of vast pasture areas (269,000 km² between 1960 and 2019 [15]), many of which are in some stage of degradation due to low investment in soil management and nutrient addition [16–19]. In these degraded areas, soil-feeding termites may play an important role by redistributing nutrients and organic matter through their mound-building activities. This redistribution could help mitigate nutrient depletion in nutrient-poor agroecosystems, although further research is needed to confirm this.

However, despite the high prevalence of *P. turricola* mounds in tropical pastures, there is still a poor understanding of their physical and chemical properties. Farmers often associate high mound density with pasture degradation, underestimating their potential benefits [20]. A deeper understanding of the physicochemical characteristics of *P. turricola* mounds would significantly enhance our knowledge of termite activity in

this region. This information would potentially contribute to developing sustainable management practices that minimize potential conflicts while taking advantage of the positive impacts of these mounds. We hypothesized that mound-building activities by soil-feeding termites improve the soil physical and chemical properties by creating island of soil fertility. To test this hypothesis, we conducted a field study to assess the content of soil macro-nutrients and soil organic C, the soil acidity, soil bulk density, total porosity, and aggregation of the termite mounds and their surrounding soils. These parameters were selected based on their importance in soil fertility and their ability to reveal the ecological impacts of termite activity in agricultural systems.

## Materials and methods

### Study area and sampling design

This study was conducted in the northwestern Colombian Amazon region, at the Centro de Investigaciones Amazónicas Macagual - César Augusto Estrada González, located at 1° 37' N and 75° 36' W, 300 m above sea level (Fig 1). The regional climate is classified as *Af* - Tropical rainforest climate (Köppen classification) with an average annual rainfall of 3793 mm, a monomodal rainfall regime with maximum precipitation distributed between April and September, an average

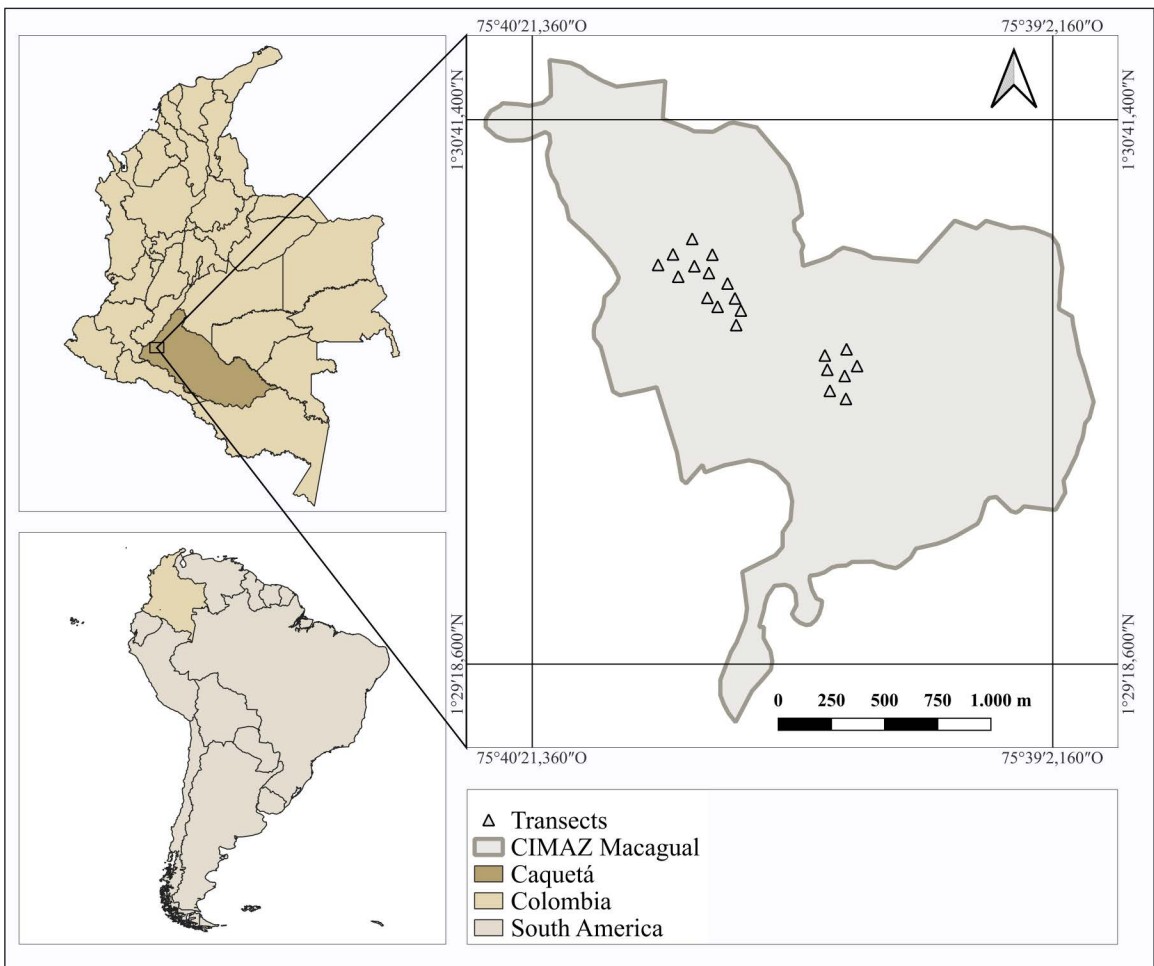

**Fig 1. Location of the study area in the Department of Caquetá, Amazon region, Colombia.** Base polygon layers for Colombia and Caquetá were sourced from the Instituto Geográfico Agustín Codazzi (IGAC) Open Database and are distributed under a CC BY 4.0 license. The South America layer was obtained from Natural Earth, a public domain dataset.

temperature of 25.5°C, and relative humidity of 84.25% [21]. The soils are classified as Ultisols according to USDA soil taxonomy [22] and are generally acidic, with high Al content that can be toxic to plants and low cation exchange capacity and base saturation, dominated by clays and with soil compaction processes reflected in high penetration resistance values [23].

To assess the creation of islands of fertility by *Patawatermes turricola*, three *Urochloa decumbes* paddocks of 1.5 ha separated by 1 km from each other, these pastures are located within the rolling hills landscape, characterized by gentle slopes. The pastures were implemented approximately 20 years ago and are characterized by a similar tree species composition, which includes scattered individuals of *Gmelina arborea* Roxb., *Erythrina poeppigiana* (Walp.) O.F.Cook, *Guarea guidonia* (L.) Sleumer, *Psidium guajava* L., *Cedrela odorata* L., *Bauhinia variegate* and L. *Cariniana pyriformis* Miers and under rotational grazing systems with a capacity of ~4 head of cattle per hectare, 5 days of occupancy and 30–35 days of recovery. In this environment, the soil-feeding termite *Patawatermes turricola* builds abundant epigeal nests of about 30 cm in height which can act as islands of soil fertility by altering the physical and chemical properties of the surrounding soil.

A total of 20 transects, each measuring 20×2 meters, were randomly established across the paddocks during April 2022. The transects were separated by an average distance of 30 meters to ensure spatial independence, on average, seven transects were placed in each pasture, with the exact number varying slightly depending on the size and configuration of the individual paddocks. Within each transect, all encountered mounds were identified, totaling 63 mounds in all study areas; 54 were determined to be *P. turricola* and were subsequently sampled. Resulting in a density of 6.75 mounds per hectare, similar to [24].

To characterize these potential islands of fertility, sampling was conducted from both the outer wall of the mounds soil and the unmodified topsoil, located at a distance of 5 meters to minimize the influence of mound-related disturbance. To evaluate soil physical properties such as aggregate stability, soil bulk density, and total porosity, undisturbed soil samples were collected using both monoliths (5×5×5 cm) and cylinders (98 cm$^3$). These samples were taken at the top of the mound (30 cm height on average) and a depth of 10 centimeters in the surrounding soil.

Disturbed soil samples were collected from the outer wall of the mound, with three replicates taken at the top, middle, and bottom sections; approximately 100 grams of soil were collected for each replicate. These replicates were combined to create composite samples representing each mound. Likewise, three disturbed soil samples were collected at 5 meters from the base of each mound to represent the surrounding habitat. These samples were combined to form a single composite sample and were taken at a depth of 10 centimeters. All composite samples were analyzed to determine the content of exchangeable cations, organic carbon, exchangeable acidity, and available phosphorus.

## Soil physical and chemical analysis

Undisturbed soil samples collected by a cylinder (98 cm$^3$) were saturated by capillarity rise and then weighed and transferred to a tension table at −6 kPa water potentials until reaching the hydraulic equilibrium, time after which the soil water content was determined in order to quantify macro-porosity, micro-porosity, and total soil porosity following the analytical methods described by [25], as well as bulk density (g cm$^{-3}$) [26].

Soil aggregate stability was determined using the Yoder wet sieving procedure [27]. For that aim, undisturbed soil blocks were used and broken along the lines of natural weakness, then passed through an 8 mm mesh. Sub-samples of 40 g were then placed on top of a set of 2, 0.25, and 0.053 mm mesh sieves for vertical oscillation (30 times per minute) in water for 30 minutes. The aggregates present on each mesh were oven-dried at 105°C for 48 hours, then weighed for calculation of the percentage of macro and micro-aggregates and the determination of the weighted mean diameter (WMD) [28,29].

Disturbed soil samples were air-dried and passed through a 2 mm sieve and used to determine soil chemical attributes related to acidity, macro-nutrients and soil organic C. The exchangeable acidity ($H^+ + Al^{3+}$) and the exchangeable aluminum ($Al^{3+}$) were extracted by KCl solution 1M and determined by titration with NaOH 0.01M using phenolphthalein as indicator

and by back titration with NaF after acidification with HCl 0.01M, respectively [22]. The content of exchangeable potassium ($K^+$), calcium ($Ca^{2+}$), Sodium ($Na^+$), and magnesium ($Mg^{2+}$) were quantified by using an atomic absorption spectrophotometer in an extract prepared with 1M ammonium acetate [30], soil base saturation (BS) was calculated as described in [31].

Plant-available phosphorus ($P_{av}$) was assessed using the Bray II method [32] through the determination of P-molybdate blue color measured on a visible spectrophotometer at 660 nm. Soil organic C (SOC) concentration was determined by a modified wet oxidation method, without an external heating procedure, followed by the colorimetric method using a UV–visible spectrophotometer at a wavelength of 645nm [33].

Termite samples were taken from each mound and deposited in 90% alcohol for subsequent identification. Identification was conducted using the key to neotropical Apicotermitinae genera based on the worker caste was used [34]. Then dissected the enteric valve using the protocol proposed by [35], and its morphology was compared to descriptions provided in existing literature to identify the species [36,37]. The authorization used for the collection of the individuals was based on resolution 01140 of 2016 issued by the National Agency of Environmental Licenses with the previous approval of the ethics and bioethics committee in research of the University of Amazonia through endorsement number 16 of March 12, 2021, approved by act 003 of 2021.

## Data analysis

The statistical analyses were conducted using R software version 4.2.1 [38]. To test the hypothesis that termite mounds create fertility islands, we compared soil parameters between unmodified topsoil samples and mound soil, each parameter was individually evaluated. Initially, the normality of the data was assessed using the Shapiro-Wilk test. Subsequently, soil parameters with normal data distribution (macro-porosity, bulk density, weighted mean diameter, and the percentage of macro and micro-aggregates) were analyzed using a Generalized Linear Model (GLM) with the "glm2" package [39].

Soil properties whose data did not follow a normal distribution were analyzed by Generalized Linear Mixed Models (GLMM) using the "lme4" package [40], negative binomial regression used for base saturation, cations, effective cation exchange capacity, exchangeable aluminum available, phosphorus, soil organic carbon and poisson regression for exchangeable acidity, macro and micro-porosity; in all models, soil type (mound soil or unmodified topsoil) was set as a fixed effect, and the transect within the paddock as a random effect. Model selection was based on the Akaike information criterion (AIC), Bayesian information criterion (BIC), and overdispersion; Fisher's multiple comparison tests ($\alpha = 0.05$) were applied using the "multcomp" package [41]. These models allowed us to evaluate the role of termite mounds in creating fertility islands by comparing soil properties between mound and surrounding soils. Finally, a Principal Component Analysis (PCA) employing a Monte-Carlo test from the "Ade4" package [42] was conducted to assess the overall impact of soil type (mound soil or unmodified topsoil) on the physicochemical properties. To facilitate visualization in the biplot graph, only variables with a contribution of 5% or more to either of the two principal components were included.

## Results

This study revealed that mound soil and unmodified topsoil have significant differences in several physicochemical properties. Mound soil exhibited notably higher levels of organic carbon, aggregate stability (reflected by the highest weighted mean diameter), and macro-porosity. Additionally, mound soil displayed elevated levels of cations and phosphorus, while adjacent topsoil had higher exchangeable acidity.

## Soil physical properties

The results obtained indicated an increase in the percentage of macro-porosity in the mound soil. Higher values of micro-porosity, total porosity (Fig 2a), and soil bulk density (Fig 2b) were observed in the unmodified topsoil.

Mound soil also showed higher values in the weight mean diameter and in the proportion of larger aggregates (>2 mm). In contrast, the unmodified topsoil presented higher values in the proportion of aggregates lower than 2 mm as well as in the percentage of micro-aggregates as detailed in Table 1.

## Soil chemical properties

The data obtained for the chemical attributes revealed significant increases in the content of $Ca^{2+}$, $Mg^{2+}$, $K^+$, $P_{av}$, SOC, and BS were found in the mound soil compared to the surrounding soil. Besides, exchangeable acidity ($H^+ + Al^{3+}$) and aluminum presented lower concentrations in the mound soil, this decrease is related to the high content of $Ca^{2+}$ and $Mg^{2+}$ in this soil (Table 2).

The PCA for the physicochemical properties explained 49.3% of the variability of the data along the first two axes (Fig 3a), clearly separating the unmodified topsoil from the mound soil (Fig 3b). Axis 1 (34.6%) opposed base saturation, organic carbon, aggregates > 2 mm, and WMD associated with the mound to soils with high aluminum content, higher micro-porosity, micro-aggregates, and high bulk density associated with unmodified topsoils. On the other hand, the axis 2 (14.7%) opposes the total porosity of the macro-aggregates. Our results show how the mounds of soil-feeding termites present better soil physicochemical characteristics.

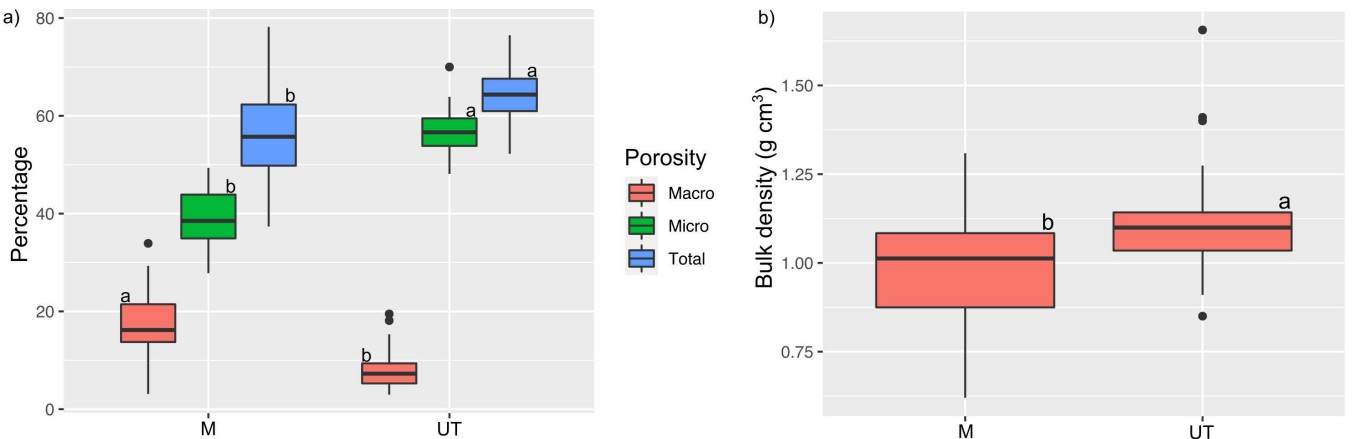

**Fig 2. Boxplots visualizing a) macro, micro, and total porosity; b) bulk density analyzed in composite soil samples of 54 mounds built by soil-feeding termites *Patawatermes turricola* (M) and the respective control topsoil (depth 0-10 cm) taken in a distance of 5 m next to each mound (UT), highlighting the formation of fertility islands.**

**Table 1. Mean ± Standard Error of physical parameters within *Patawatermes turricola* fertility islands (Mound Soil) and surrounding (Unmodified Topsoil 0-10 cm) in Colombian Amazonian pastures (n = 54). Different letters within the same row indicate significant differences (p < 0.05) as determined by Fisher's LSD test, highlighting the impact of termite mound-building on soil properties.**

| Parameter | Mound Soil | Unmodified Topsoil |
|---|---|---|
| | Mean ± S.E | Mean ± S.E |
| Aggregates >0.053–0.250 mm (%) | 0.74 ± 0.114 b | 1.16 ± 0.092 a |
| Aggregates >0.250–2 mm (%) | 19.5 ± 1.27 b | 22.7 ±1.27 a |
| Aggregates >2 mm (%) | 74.9 ± 1.86 a | 68.5 ±1.86 b |
| Macro-aggregates (%) | 92.15 ± 5.55 a | 88.88 ± 5.74 a |
| Micro-aggregates (%) | 0.71 ± 0.12 b | 1.17 ± 0.10 a |
| Weighted mean diameter (mm) | 3.88 ± 0.07 a | 3.57 ± 0.07 b |

**Table 2. Mean ± Standard Error of chemical parameters within *Patawatermes turricola* fertility islands (Mound Soil) and surrounding (Unmodified Topsoil 0-10 cm) in Colombian Amazonian pastures (n = 54). Different lowercase letters within the same row indicate significant differences (p < 0.05) as determined by Fisher's LSD test, illustrating the chemical changes associated with termite mound-driven fertility islands. $Al^{3+}$: exchangeable aluminum, $H+Al^{3+}$: exchangeable acidity, BS: base saturation, $Ca^{2+}$: Calcium, ECEC: effective cation exchange capacity, $K^+$: Potassium, $Mg^{2+}$: Magnesium, $Na^+$: Sodium, Pav: available phosphorus, SOC: soil organic carbon.**

| Parameter | Mound Soil | Unmodified Topsoil |
|---|---|---|
| | Mean ± S.E | Mean ± S.E |
| $(Al^{3+})$ (mmol $kg^{-1}$) | 14.02 ±4.73 b | 20.61 ± 6.93 b |
| $(H^+ + Al^{3+})$ (mmol $kg^{-1}$) | 6.31±2.15 b | 15.91 ±5.37 a |
| BS (%) | 73.0 ± 24.40 a | 50.68 ± 16.95b |
| $Ca^{2+}$ (mmol $kg^{-1}$) | 17.98 ± 6.06 a | 13.95 ± 4.71 b |
| ECEC (mmol $kg^{-1}$) | 43.19 ± 14.46 a | 40.15 ± 13.44 a |
| $K^+$ (mmol $kg^{-1}$) | 6.84 ± 2.34 a | 2.96 ± 1.04 b |
| $Mg^{2+}$ (mmol $kg^{-1}$) | 11.03 ± 3.74 a | 5.26 ± 1.80 b |
| $Na^+$ (mmol $kg^{-1}$) | 0.84 ± 0.33 a | 0.73 ± 0.29 a |
| $P_{av}$ (mg $kg^{-1}$) | 15.92 ± 5.39 a | 7.65 ± 2.61 b |
| SOC (g $kg^{-1}$) | 15.92 ± 5.39 a | 7.65 ± 2.61 b |

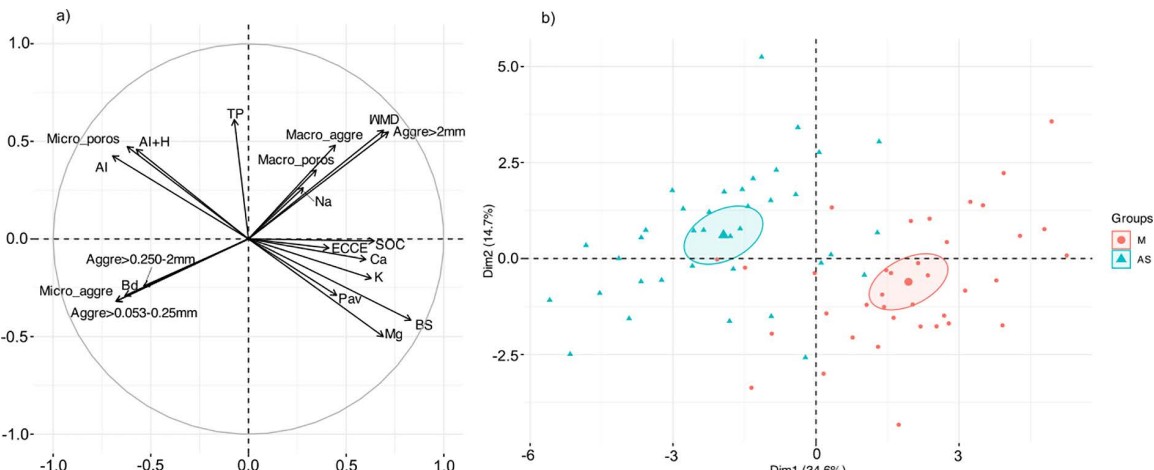

**Fig 3. Principal component analysis biplot projected onto the F1/F2 plane and the grouping of sites by soil type. Observation: 0.22; p-value: 0.001. M: Mound Soil, UT: Unmodified Topsoil.** Aggre>0.053–0.25mm: Aggregates >0.053–0.250 mm (%), Aggre>0.250–2mm: Aggregates >0.250–2 mm (%), Aggre>2mm: Aggregates >2 mm (%), Al: exchangeable aluminum (mmol $kg^{-1}$), Al+H: exchangeable acidity (mmol $kg^{-1}$), Bd: Bulk density (g $cm^3$), BS: base saturation (%), Ca: Calcium (mmol $kg^{-1}$), ECEC: effective cation exchange capacity (mmol $kg^{-1}$), K: Potassium (mmol $kg^{-1}$), Mg: Magnesium (mmol $kg^{-1}$), Macro_aggre: Macro-aggregates (%), Micro_aggre: Micro-aggregates (%), Macro_poros: Macro-porosity (%), Mico_poros: Micro-porosity (%), Na: Sodium (mmol $kg^{-1}$), Pav: available phosphorus (mg $kg^{-1}$), SOC: soil organic carbon (g $kg^{-1}$), TP: total porosity, WMD: weighted mean diameter (mm).The contribution of each parameter to the formation of the principal components is represented by a color gradient from blue (low contribution) to red (high contribution).

## Discussion

The findings reveal a significant improvement in soil fertility within the mound soil, characterized by enhancements in both chemical and physical properties. This fertility directly benefits pasture growth on the mounds, where the accumulation of organic matter and minerals creates a more favorable environment for vegetation. While the surrounding soil may be nutrient poor, these mounds act as fertility hotspots, improving the soil structure and nutrient availability, which in turn promote better conditions for the growth of pasture.

The crucial role of termite activity in driving these ecological processes offers a promising way to improve grassland productivity and ecosystem sustainability in tropical landscapes. Since termite activity appears to have an impact that extends beyond the mound, suggesting a long-term effect on vegetation patterns and ecosystem dynamics, like an extended phenotype [3]. This ability to create patchy vegetation mosaics further highlights the importance of termites in shaping the ecological landscape.

## Effect of soil-feeding termites on soil physical properties in tropical pastures

The results indicate a lower percentage of macro-porosity and higher micro-porosity in the adjacent topsoil, suggesting greater compaction, likely due to cattle trampling, as previously documented in livestock systems within the Colombian Amazon [43,44]. In contrast, the mound soil exhibited reduced compaction, evidenced by increased macro-porosity and a decrease in micro-porosity. Although recent studies [45,46] have reported higher micro-porosity in termite mounds, these findings may be influenced by variations in habitat types, soil characteristics, and termite feeding groups. Consequently, direct comparisons may have certain limitations.

Some authors have shown that termite mounds exhibit increased macro-porosity [47], a consequence of termites transporting and redistributing subsoil materials to the surface for mound construction. This observation raises the possibility that termite activity primarily involves the relocation of existing resources, rather than the creation of new ones. However, our findings also demonstrate that termite activity leads to significant improvements in soil physical structure, implying that even if resources are merely redistributed, the resulting changes can enhance soil function and fertility. The extent of these physical modifications may be contingent upon soil texture [7], as evidenced by a study on Acrisols where high clay content, in conjunction with termite activity, resulted in cementation and compaction, leading to increased bulk density within the mounds [48].

The analyses also revealed greater structural stability in mound soil compared to surrounding areas. This was evident in both higher weighted mean diameter of aggregates and their specific proportions, which aligns with increased porosity observed earlier. Such enhanced stability might be linked to the study's soil type, as [13] suggest higher clay content in tropical soils promotes increased mound stability. Our findings are consistent with previous studies [49,50] demonstrating similar trends, possibly due to termite-related processes like the creation of cementing particles during their feeding activities.

The higher proportion of large aggregates (>2mm) and lower proportion of smaller ones (<0.053mm) compared to surrounding soils aligns with observations by [12] linking larger aggregates created by soil-feeding termites to increased structural stability. However, the variability in soil type, land use, and termite feeding habits reported in other studies [51,52] suggests that these findings not be universally applicable, and therefore, further research is needed to elucidate the specific factors influencing aggregate size distribution in termite-modified soils.

## Termite mounds as islands of soil chemical fertility in tropical pastures

The mound soil displayed markedly different chemical characteristics than the surrounding soil, notably exhibiting a substantial enrichment in nutrients and organic carbon. This enrichment is likely a result of termite feeding and nest-building activities, effectively transforming termite mounds into 'chemical fertility islands' within the landscape [53]. Nevertheless, the possibility that this enrichment reflects a redistribution of nutrients from surrounding soils rather than a net gain should be considered. While the observed increase in nutrient levels within the mounds suggests a positive impact on soil fertility, further investigation is warranted to quantify the degree of resource redistribution and its subsequent effects on the fertility of the surrounding soil matrix.

A recent review [54] highlights that carbon, phosphorus, and potassium levels are consistently higher in termite-modified soils compared to bare soils. This nutrient enrichment is primarily attributed to termite activities, including the extraction of nutrients from ingested materials and the formation of organic compounds through biochemical processes

within the termite digestive tract. Additionally, the intestinal transit of soil-feeding termites plays a crucial role in stabilizing soil organic matter (SOM) and protecting it from rapid turnover, a common occurrence in tropical environments [55]. This stabilization process enhances the long-term persistence of organic matter in termite-modified soils, contributing to their increased fertility.

The higher exchangeable cations in mound soil than in the adjacent topsoil is consistent with previous studies [9,13]. This enrichment is attributed to various mechanisms, including using organic matter and clay by termites for nest construction, decomposition of organic matter, and transporting cation-rich clay from the subsoil for nest construction [47,53,56]. As a result, mound soils have higher cation exchange capacity and base saturation [56]. Similarly, the termite activity resulted in a significant increase in phosphorus content within the mound soil. Given the critical role of phosphorus as a macro-nutrient and its frequent deficiency in highly weathered tropical soils, this finding is particularly significant. This improvement in soil fertility has important implications for agroecosystems [57], as erosion and human activities can redistribute mound material, gradually improving the fertility of the surrounding soil [58]. While the mound may initially serve as a localized point of nutrient enrichment, the long-term effects of termite-mediated nutrient cycling could contribute to broader improvements in soil fertility and agricultural productivity [59].

The increase in phosphorus content within the mound soil can be explained by two mechanisms associated with the following process: i) the alkaline environment of the termite digestive system (pH up to 12.5) can release phosphorus from soil aluminum [60], and ii) enrichment of biological origin since phosphorus is added to the mound through termite saliva, feces, and plant remains used in nest construction [55,61]. Furthermore, this last mechanism is also related to the increase in organic carbon content in termite mounds. Fecal material and saliva adhere to soil particles during nest construction and enrich the soil with carbon [62, 63], improving the soil aggregate stability in mound soil. Our results are consistent with those of [7,9], who found higher organic carbon content in mound soil. They attributed it to granulation during nest construction through soil ingestion and selecting organic particles to return them in oral and fecal granules.

Furthermore, some authors also point out that the plant material used in constructing termite mounds can contribute to carbon increases in the mounds [61]. In contrast, other authors report lower organic carbon content in mound soil, possibly due to the type of soil poor in organic matter in the deeper layers used in mound construction and to differences in termite feeding habits [64,65]. Reduction in soil exchangeable acidity in mounds in contrast to the surrounding soil is consistent with the findings of [9] in degraded grassland transitioning to secondary forest. This result could be related to the high content of Ca and Mg in termite mound soil. In addition, since soil acidity is inversely related to pH, the low exchangeable acidity in the mound soil may be influenced by the transit of soil through the termite's digestive tract, which has an alkaline medium [7,66].

This study is of particular importance in the context of the Amazon region, where soils are characterized by their acidic nature, low nutrient content and compacted structure, the findings show that termite feeding and nest-building activity significantly modified the physicochemical properties of the soil, resulting in less soil compaction and greater soil structural stability, accompanied by a considerable increase in nutrients and a decrease in acidity. These positive changes suggest that termites play a crucial role in maintaining soil health within degraded Amazonian pastures.

This study demonstrated that the soil-feeding termite *Patawatermes turricola* significantly modifies the physical and chemical properties of soils through its feeding and nest-building activities. Mound soil exhibited improved porosity, bulk density, aggregate stability, and nutrient content—key indicators of soil fertility. These changes are likely attributed to soil transfer for biogenic structure construction, termite feeding activity, and the addition of fecal and buccal granules. While mound nutrient enrichment was evident, discerning between a net gain and resource redistribution necessitates further investigation. Recognizing this alternative perspective strengthens our interpretations and underscores the complexity of termite-mediated soil processes. Although these fertility attributes were predominantly observed within mounds, the potential for gradually enhancing surrounding degraded pastures through mound material redistribution via erosion and termite activity exists.

However, our findings represent a snapshot in time, and long-term monitoring is crucial to fully assess the persistence and ecological implications of these modifications. Additionally, our study focused on a specific location, limiting the generalizability of our results. Future research across diverse soil types, land-use histories, and termite species is required to broaden our understanding. Specifically, efforts should prioritize quantifying nutrient depletion around mounds, examining mound turnover and erosion, and conducting detailed soil profile analyses. These initiatives refine our comprehension of termite impacts on tropical soil fertility, clarifying the mechanisms through which termite activity may enhance agroecosystems broadly, extending beyond immediate mound areas.

## Acknowledgments

We are grateful to members of Semillero de Investigación de Biología del Suelo for their support during field trips.

## Author contributions

**Conceptualization:** Ervin Humprey Duran-Bautista.

**Data curation:** Ervin Humprey Duran-Bautista, Adriana M. Silva-Olaya, María Paula Llanos-Cabrera, Katherin Yalanda-Sepúlveda, Juan Carlos Suárez.

**Formal analysis:** Ervin Humprey Duran-Bautista, Adriana M. Silva-Olaya, María Paula Llanos-Cabrera, Katherin Yalanda-Sepúlveda, Juan Carlos Suárez.

**Funding acquisition:** Ervin Humprey Duran-Bautista.

**Investigation:** Ervin Humprey Duran-Bautista, Adriana M. Silva-Olaya.

**Writing – original draft:** Ervin Humprey Duran-Bautista, Adriana M. Silva-Olaya, María Paula Llanos-Cabrera, Katherin Yalanda-Sepúlveda, Juan Carlos Suárez.

**Writing – review & editing:** Ervin Humprey Duran-Bautista, Adriana M. Silva-Olaya, María Paula Llanos-Cabrera, Katherin Yalanda-Sepúlveda, Juan Carlos Suárez.

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
