## [Decision Letter · Decision Letter 0]

27 Aug 2024

PONE-D-24-16884Soil-feeding termites build islands of soil physical and chemical fertility in pastures in Colombian AmazonPLOS ONE

Dear Dr. Duran-Bautista,

Thank you for submitting your manuscript to PLOS ONE. After careful consideration, we feel that it has merit but does not fully meet PLOS ONE’s publication criteria as it currently stands. Therefore, we invite you to submit a revised version of the manuscript that addresses the points raised during the review process.

We look forward to receiving your revised manuscript.

Kind regards,

Kamlesh Jangid, Ph.D

Academic Editor

PLOS ONE

Journal Requirements:

"This work was funded by Universidad de la Amazonia through project 600.6.6331"

5. Please provide a complete Data Availability Statement in the submission form, ensuring you include all necessary access information or a reason for why you are unable to make your data freely accessible. If your research concerns only data provided within your submission, please write "All data are in the manuscript and/or supporting information files" as your Data Availability Statement.

6. One of the noted authors is a group or consortium: Universidad de la Amazonia / Consorcio Colombia

In addition to naming the author group, please list the individual authors and affiliations within this group in the acknowledgments section of your manuscript. Please also indicate clearly a lead author for this group along with a contact email address.

8. We note that Figure 1 in your submission contain map images which may be copyrighted. All PLOS content is published under the Creative Commons Attribution License (CC BY 4.0), which means that the manuscript, images, and Supporting Information files will be freely available online, and any third party is permitted to access, download, copy, distribute, and use these materials in any way, even commercially, with proper attribution. For these reasons, we cannot publish previously copyrighted maps or satellite images created using proprietary data, such as Google software (Google Maps, Street View, and Earth). For more information, see our copyright guidelines: http://journals.plos.org/plosone/s/licenses-and-copyright.

1) You may seek permission from the original copyright holder of Figure 1 to publish the content specifically under the CC BY 4.0 license.  

2) If you are unable to obtain permission from the original copyright holder to publish these figures under the CC BY 4.0 license or if the copyright holder’s requirements are incompatible with the CC BY 4.0 license, please either i) remove the figure or ii) supply a replacement figure that complies with the CC BY 4.0 license. Please check copyright information on all replacement figures and update the figure caption with source information. If applicable, please specify in the figure caption text when a figure is similar but not identical to the original image and is therefore for illustrative purposes only.

**Additional Editor Comments:**

Dear Prof. Ervin Humprey Duran-Bautista,

Thank you for your submission to PLoS One. I have read your manuscript with great interested and found it within the scope of the journal. I have also received reports from the two experts, and they are appended here (one in the online system and the other in-line below) for your reference. Collectively, I and of the opinion that the manuscript requires a major revision before we can consider it further. I am therefore returning it to you accordingly.

As you'll notice, both reviewers highlight several areas of concern which need your careful attention. Several experimental details are either missing or seems unconventional, especially the sample collected from the "adjacent soils", these need to be corrected for better accuracy and to ensure the validity of your results. Details on statistical analysis seem confusing as well. I personally feel that although you have interesting results, they are not appropriately laid out and discussed. These two sections require rearrangement, as recommended by the Rev#1. In the discussion, the implications of your work in habitat management or conservation efforts are very speculative and this needs to be dealt with more direct approach and accuracy.

Overall, the manuscript requires a significant improvement before it can be considered for another review. I hope you'll find these suggestions useful to improve it and resubmit.

With Kind regards,

Kamlesh Jangid, PhD

Academic Editor, PLoS One

Detailed Reviewer Comments Follow:

Reviewer #1: See file in the system.

Reviewer #2: See below

1. Is the manuscript technically sound, and do the data support the conclusions?

Partly.

The study clearly demonstrates the physical and chemical differences between the mound and the adjacent soil, allowing termites to be characterized as builders of fertility islands in degraded pastures. I understand that this is a strong point of the study. However, the study does not demonstrate how these differences can contribute to the pastoral system, or how grazing could benefit from the presence of mounds. Therefore, improving the development of the discussion of the results would be opportune to elucidate what the positive impact indicated in the conclusion would be.

In this sense, some specific observations that could be improved by the authors are listed below:

Lines 72-76: The authors' hypothesis mentions that the best conditions of the mound's chemical and physical properties may be important for nutrient-poor agroecosystems. Given this, it is essential that the authors highlight this importance for the agroecosystem in the discussion of the results. I believe that this may favor a stronger conclusion.

Lines 97-98: It is mentioned that mounds are abundant in these environments, however, in the following paragraph it is not clear whether all mounds within the transects were studied or only some of them. Furthermore, the abundance of these mounds could compete with the useful pasture area, constituting a likely negative impact by reducing pasture area.

Lines 177-178: The mention of "better" characteristics here seems to refer to agronomic aspects. However, it is not clear how the pasture could benefit from the fertility of the mound, considering that the pasture grows on adjacent soil that remains poor in nutrients. This argument needs to be developed in the discussion.

Lines 228-230: Here the importance of termites is mentioned superficially for the conservationist management of tropical pastures. I suggest discussing in a concrete way the potential role of termites in ecological processes and how this could favor conservationist management or better pasture conditions. The aspect that seems important is that even with fertility in the mound, it is not demonstrated how this could favor a better condition of the agroecosystem.

Lines 268-271: Could it be possible that termites represent a reduction in organic matter in the adjacent soil and consequently a reduction in some chemical fertility attributes? If this were the case, termites would actually be causing a negative impact, accumulating soil fertility for themselves. I suggest developing the discussion to support this possibility. Perhaps the hypothesis of a largely positive impact of termite activity is not fully supported by the data.

Lines 279-281: Here the higher level of phosphorus available in the mound is mentioned as important for tropical soils. This result deserves to be highlighted, but it is important to demonstrate whether this can contribute to the agroecosystem, or whether it is restricted to the mound.

Lines 310-312. Fertility attributes were higher only in mounds. Suggesting that termite activity contributes to improving the conditions of degraded pastures needs more support from the data and discussion of the results. I suggest revising and improving this aspect of the discussion to strengthen a more robust conclusion.

2. Has the statistical analysis been performed appropriately and rigorously?

No

The statistical methods seem adequate. However, the data in Table 2 need to be revised regarding the unit. The authors should check whether the mmol unit of each attribute is correct, or should it be cmol. In addition, the calculations of base saturation and cation exchange capacity are inconsistent. It is not reasonable, for example, that the ECEC is lower than the Ca2+ content. Therefore, Table 2 needs to be revised. Consequently, the authors should check whether these data used in the principal component analysis also need to be revised.

Figures 2 and 3 are in reverse order and do not correspond exactly as presented.

3. Have the authors made all data underlying the findings in their manuscript fully available?

No

Authors declare restrictions.

4. Is the manuscript presented in an intelligible fashion and written in standard English?

No

The manuscript could be improved by reviewing the English language and using uniform terms (e.g. macro-porosity; macroporosity and macro-pores; microagregados and microaggregates; Mico_poros, micro-porosity and microporosity).

Reviewers' comments:

Reviewer's Responses to Questions

**Comments to the Author**

1. Is the manuscript technically sound, and do the data support the conclusions?

Reviewer #1: Yes

Reviewer #2: Partly

2. Has the statistical analysis been performed appropriately and rigorously? 

Reviewer #1: Yes

Reviewer #2: No

3. Have the authors made all data underlying the findings in their manuscript fully available?

Reviewer #1: No

Reviewer #2: No

4. Is the manuscript presented in an intelligible fashion and written in standard English?

Reviewer #1: No

Reviewer #2: No

5. Review Comments to the Author

Reviewer #1: Please refer to the uploaded document "20240709_Review_Soil-feeding termites build islands of soil physical and chemical fertility in pastures in Colombian Amazon.docx." This document contains detailed comments and suggestions regarding the manuscript, including an evaluation of the study's methodology, results, and discussion, as well as specific questions and recommendations for the authors to consider in their revisions.

Reviewer #2: 1. Is the manuscript technically sound, and do the data support the conclusions?

Partly.

The study clearly demonstrates the physical and chemical differences between the mound and the adjacent soil, allowing termites to be characterized as builders of fertility islands in degraded pastures. I understand that this is a strong point of the study. However, the study does not demonstrate how these differences can contribute to the pastoral system, or how grazing could benefit from the presence of mounds. Therefore, improving the development of the discussion of the results would be opportune to elucidate what the positive impact indicated in the conclusion would be.

In this sense, some specific observations that could be improved by the authors are listed below:

Lines 72-76: The authors' hypothesis mentions that the best conditions of the mound's chemical and physical properties may be important for nutrient-poor agroecosystems. Given this, it is essential that the authors highlight this importance for the agroecosystem in the discussion of the results. I believe that this may favor a stronger conclusion.

Lines 97-98: It is mentioned that mounds are abundant in these environments, however, in the following paragraph it is not clear whether all mounds within the transects were studied or only some of them. Furthermore, the abundance of these mounds could compete with the useful pasture area, constituting a likely negative impact by reducing pasture area.

Lines 177-178: The mention of "better" characteristics here seems to refer to agronomic aspects. However, it is not clear how the pasture could benefit from the fertility of the mound, considering that the pasture grows on adjacent soil that remains poor in nutrients. This argument needs to be developed in the discussion.

Lines 228-230: Here the importance of termites is mentioned superficially for the conservationist management of tropical pastures. I suggest discussing in a concrete way the potential role of termites in ecological processes and how this could favor conservationist management or better pasture conditions. The aspect that seems important is that even with fertility in the mound, it is not demonstrated how this could favor a better condition of the agroecosystem.

Lines 268-271: Could it be possible that termites represent a reduction in organic matter in the adjacent soil and consequently a reduction in some chemical fertility attributes? If this were the case, termites would actually be causing a negative impact, accumulating soil fertility for themselves. I suggest developing the discussion to support this possibility. Perhaps the hypothesis of a largely positive impact of termite activity is not fully supported by the data.

Lines 279-281: Here the higher level of phosphorus available in the mound is mentioned as important for tropical soils. This result deserves to be highlighted, but it is important to demonstrate whether this can contribute to the agroecosystem, or whether it is restricted to the mound.

Lines 310-312. Fertility attributes were higher only in mounds. Suggesting that termite activity contributes to improving the conditions of degraded pastures needs more support from the data and discussion of the results. I suggest revising and improving this aspect of the discussion to strengthen a more robust conclusion.

2. Has the statistical analysis been performed appropriately and rigorously?

No

The statistical methods seem adequate. However, the data in Table 2 need to be revised regarding the unit. The authors should check whether the mmol unit of each attribute is correct, or should it be cmol. In addition, the calculations of base saturation and cation exchange capacity are inconsistent. It is not reasonable, for example, that the ECEC is lower than the Ca2+ content. Therefore, Table 2 needs to be revised. Consequently, the authors should check whether these data used in the principal component analysis also need to be revised.

Figures 2 and 3 are in reverse order and do not correspond exactly as presented.

3. Have the authors made all data underlying the findings in their manuscript fully available?

No

Authors declare restrictions.

4. Is the manuscript presented in an intelligible fashion and written in standard English?

No

The manuscript could be improved by reviewing the English language and using uniform terms (e.g. macro-porosity; macroporosity and macro-pores; microagregados and microaggregates; Mico_poros, micro-porosity and microporosity).

6. PLOS authors have the option to publish the peer review history of their article (what does this mean? ). If published, this will include your full peer review and any attached files.

**Do you want your identity to be public for this peer review?** For information about this choice, including consent withdrawal, please see our Privacy Policy .

Reviewer #1: No

Reviewer #2: No

---

## [Author Response · Author response to Decision Letter 0]

9 Nov 2024

Response to Reviewers’ Comments on PONE-D-24-16884

Ms. Ref. No.: PONE-D-24-16884

Soil-feeding termites build islands of soil physical and chemical fertility in pastures in Colombian Amazon

PLOS ONE

Attention:

Kamlesh Jangid, PhD

Academic Editor, PLoS One

PLOS ONE

Dear Dr Kamlesh Jangid

Many thanks for forwarding to us the comments from the two reviewers. We found the comments of the reviewers to be objective, constructive and helpful, and we have done our best to employ them wherever possible for the improvement of the manuscript. The authors have declared that no competing interests exist.

We describe below our response to comments made by reviewers. Standard font is used for the comments, while red font is used for our response to comments.

Specific responses to the Editorial Board Member and reviewers’ concerns: Our responses are in red font.

Reviewer 1

Summary of the research

The study was conducted in three pastures located in the northwestern Colombian Amazon, characterized by a mean annual precipitation of about 3700 mm. The soils are leached, nutrient deficient, acidic, and compacted red Ultisols with aluminum concentrations potentially toxic to plant growth. The study addresses the important topic of soil-organism interaction. The authors investigated how physicochemical properties of epigeal mounds built by the soil-feeding termite species Patawatermes turricola differ from termite-unmodified reference soil (habitat topsoil). Physical soil parameters assessed included aggregate stability, macro- and microporosity, and bulk density. Chemical parameters included soil organic carbon, exchangeable acidity, aluminum toxicity, cation exchange capacity, base saturation, and concentrations of macronutrients (plant-available phosphorus, Mg, Ca, K, Na). These parameters are indicators of soil health, soil fertility, toxicity, soil structure, that is parameters crucial for plant establishment and development. Previous studies mostly investigated physicochemical parameters of epigeal mounds built by fungus-growing termites in semi-arid agroecosystems and savannas in Africa, where fungus-growing termites are the main year-round active decomposers and soil bioturbators. In contrast, unfortunately, there is limited knowledge about the characteristics of epigeal mounds built by neotropical soil-feeding termites, and their role as ecosystem engineers in agroecosystems of the New World tropics.

The topic addressed in the article clearly lies in the scope of the journal PLOS One. The sample size of 54 mound soil and control soil samples is sufficient (and the methods used are appropriate) to draw robust conclusions about the characteristics of P. turricola mounds in the study sites.

However, I found some important information lacking, such as how the P. turricola mounds could improve soil health, alongside details on mound densities, mound soil quantities, turnover/erosion rate. It would be beneficial to explain why P. turricola mounds and the analyzed physicochemical parameters were selected. Additionally, it would be informative to know if there are other epigeal-mound builders in the region, whether soil-feeding termites are the primary decomposers/ bioturbators, and the presence and impact of other soil macrofauna, particular earthworms, given the region's high annual precipitation.

These points should be addressed to strengthen the manuscript and provide a comprehensive understanding of the study's findings and implications. Most of this information should be readily available. If data on mound soil quantity are lacking, this could be acknowledged and discussed without impacting the manuscript's acceptance. I suggest a thorough revision of the Discussion section, ensuring the terms used (e.g., macro- and microporosity) align with their definitions in the cited literature (see comments below).

Answer: We are very grateful for each of the comments made, which we have considered very carefully. Your comments were very important for the improvement of the manuscript.

Additionally, the manuscript would benefit from English language editing.

Answer: We have carefully checked and adjusted the language of the manuscript.

Overall, the article is suitable for publication pending revisions addressing both minor and major points. Specific comments and questions for the authors are provided below, unfortunately not differentiated into major and minor points - for which I apologize.

Answer: Once again, we thank you for your comments, which we value very highly.

Soil-feeding termites build islands of soil physical and chemical fertility in pastures in Colombian Amazon

Remark: Abbreviate the genus after first introduction when referring to termite species (P. turricola)

Answer: We have made the adjustment throughout the manuscript.

Abstract:

Line 20: … feeding and nesting activities and the construction of biogenic structures …E.g. Line 21ff: State early in the abstract that the mounds were built by soil-feeding termite species. For example, "mounds constructed by the soil-feeding termite species Patawatermes turricola…” (Why this species? Is P. turricola the predominant termite species/most conspicuous, most widespread mound-builde/macrofauna?

Answer: We have made the suggested adjustment indicating that this species is abundant in the region.

… some useful information about study site would be habitat type (pasture), region (northwestern Colombian Amazon), mean annual precipitation about 3800 mm, - soil type Ultisol, characterized by - leached, nutrient deficient, .. (most prominent parameters potentially influenced by soil-feeding termites).

Briefly state your hypothesis and, in the last sentence, relate the findings to your hypothesis.

Why did you select/study these physicochemical characteristics of these mounds? (Example: due to trampling of animals, the pastures have especially compacted soils? Therefore aggregate stability and macro/microporosity of mound soil?

Define " better values/improvement" in your context: beneficial for plant growth, other macrofauna, soil structure? How can mound properties positively influence the habitat? Via erosion of mound-soil, turn-over? As stated by Sarcinelli et al. (2013) “With time and abandonment, termite mounds are eroded, and their material is redistributed on the soil surface, potentially creating a soil environment more favorable to plant establishment and development.”

How can findings be useful for habitat management or conservation?

Answer: We have made the suggested adjustments

Keywords: “bioturbation” – Soil bioturbation is not directly addressed or quantified in the study.

Information about P. turricola mounds in the manuscript: 30 cm high – Was the basal diameter/circumference assessed? Can you roughly estimate the amount of soil bioturbated to construct these mounds? This could contribute information on bioturbation during mound construction.

Answer: We have changed the keyword bioturbation to mounds which better fits the scope of the study.

INTRODUCTION

Lines 47ff: The authors describe soil-feeding termites as feeding group. It would be good to very briefly name which other feeding groups do exist (or are important in the study region). For instance referring to the classification of Donovan – the authors last reference [62]: (Donovan et al., 2001)

🡪 as well because of Line 56:… the magnitude of these differences depends on feeding habits, the soil properties, the termite species, mound age, vegetation, and land use..”

… “This fecal building process results in unique physico-chemical properties of these termite mounds, which are characterized by higher moisture and higher concentration of organic matter and cations.”

→ Are these mound parameters characteristic for soil-feeder mounds? Is higher moisture content in mounds also important in region with 3900 mm mean annual rainfall?

Are earthworms important in the study region/sites as well? (termites known to be dominant decomposers and soil bioturbators especially in dry, degraded habitats – as well in your sites?)

Answer: We have incorporated the suggested changes to address the questions raised However, we did not include information regarding earthworms, as we found insufficient evidence of their presence at the study site.

Line 61ff: “…therefore playing an important role in soil fertility. This is especially true in the Amazon basin, where extensive livestock production has led to the establishment of vast pasture areas (269,000 km² between 1960 and 2019, [15]), which are in some stage of degradation due to the low investment in soil management and nutrient addition”

🡪 What do you expect as most important effects of the selected soil-feeders and their mounds in the study region/study sites?

Answer: We have included new information to answer the question

Line 66: “However, despite the high prevalence of epigeous termite mounds in tropical pastures..”

🡪 Is the density of P. turricola mounds high?

Lines 67f: “…being under-appreciated by farmers who commonly associate high mound density with pasture degradation”

🡪 what do the farmers do? What would you want farmers to appreciate? Are there plans to convert the pastures into farmland? Why is it important to know the mounds physicochemical characteristics?

Lines 72ff: “We hypothesized that mound-building activities by soil-feeding termites improve the soil physical and chemical properties by creating island of soil fertility that can be particularly important in those nutrient-poor agroecosystems. To test this hypothesis, we conducted a field study to assess the content of soil macronutrients and soil organic C, the soil acidity, soil bulk density, porosity and aggregation of the termite mounds and their surrounding soils.”

Questions: Did your expectations on the effect of soil-feeding termite mounds in the study sites influence the soil parameters selected for your study? Are the selected parameters important for soils present in your area? For example see comment to Lines 86f (aluminum toxicity).

🡪 Short reasoning of parameter selection, e.g. aggregate stability: important physical indicator of soil health, protects organic matter accumulation, improves soil porosity, drainage and water availability for plants, decreases soil compaction, supports biological activity, and nutrient cycling in the soil.

Answer: We have included new information to answer the question

MATERIALS AND METHODS

Study area and sampling design

Question: Do you have photos showing P. turricola mounds in your study sites?

Line 84ff: “The regional climate is classified as AF type (Köppen classification) with an average annual rainfall of 3793 mm…”

-- The type is “Af” (second letter lowercase) and maybe add meaning i.e. Af=”tropical rainforest climate or equatorial climate”… information regarding rainfall distribution, e.g. …mono-modal rainfall distribution, with maximum precipitation between the months of ?? and ??.

Study period? When (year, months) were soil samples taken?

Answer: We have included new information to answer the question

Line 86f: …Ultisols… - Are soluble Al3+ concentrations in a range that is toxic to plants? If so, perhaps "Al toxicity" would be more appropriate than "high Al content." Sarcinelli et al. (2013) stated in their review that aluminum toxicity was apparently reduced by the mound-building activity of termites. This provides an argument for analyzing available Al3+ concentrations in the present study.

Answer: We have included new information to answer the question

Lines 93ff:

What are (roughly) the sizes of the three pastures studied and how far are they separated from each other? Did they differ in altitude, topography, site history, or any other characteristics? What were criteria to select these three pastures?

And: In the section Data analysis (GLM, GLMM) it is written that plot was set as a random effect. Does “plot” refer to the three pastures?

Do all epigeal termite mounds in the study sites belong to P. turricola? What is the shape of the mounds - conical? In their study, Duran-Bautista et al. (2023) assessed epigeal termite mounds in pastures which seem to be pretty close to the sites in the present study. There, a second soil-feeding termite built mounds although in lower densities, and the P. turricola mounds seem to be somewhat smaller.

Answer: We have included new information to provide clarity to the questions raised.

Line 100: “A total of 20 transects, each measuring 20x2 meters, were established across the paddocks.” – How did you select the locations and the directions of the 20 belt-transects?

What was the distance between the transects? How many transects placed in each pasture?

Answer: We have included new information to answer the question

Line 101: “…54 mounds of Patawatermes turricola were carefully assessed.” Which parameters have been assessed? Was the basal area /circumference and height of each mound assessed? Can you estimate the amount of mound soil?

Can you estimate the density of P. turricola mounds in your study sites? This information is better comparable with other areas and all the necessary information seem to be available:

20 belt-transects of 20x2m assessed = 800 m² 🡪 54 P. turricola mounds in 800 m² 🡪 Density: 6.75 mounds in 100m² or per hectare (or whatever area is appropriate to state) .

Answer: We have included the requested information

Line 101ff: Soil samples taken from the mound: monoliths and cylinders at height of 30cm: How did you take these samples in a height of 30 cm from the base if mounds are 30 cm tall? Did you have enough space in between these two undisturbed samples?

Was there a reason to take the habitat soil samples in a distance of 5 m? Why assumed to be unmodified by termite action/ not influenced by mound erosion? “at depth of 10cm” means samples taken in topsoil horizon 0-10cm, right?

Line 108ff: Three disturbed soil samples of mound: top, middle, bottom 🡪 from the outer mound wall, right? How much soil did you (roughly) take each to form the composite samples? The disturbed control soil samples taken in a distance of 5 m next to the mound: as well three replicates taken for composite samples of habitat topsoil? What was the distance between these three topsoil samples/selection of soil sampling location?

Answer: We have included new information to answer the question

General remark:

I found it quite difficult to refer to the topsoil samples taken in a distance of 5 m next to each mound as "adjacent soil". Another term might be easier - e.g. "(recently) unmodified habitat soil", "unmodified topsoil", "control soil", “adjacent, termite-unmodified control soils”, “termite-unmodified reference soil”... whatever term seems most appropriate to indicate that when compared to the control topsoil of the habitat, the differences in physicochemical parameters are due to the soil modification by termites during mound construction.

Answer: In accordance with the suggestion, we have decided to modify the “adjacent soil” by “unmodified topsoil”

Data analysis

Lines 153ff:

To facilitate citations for the packages used in R (some references are provided), it might be easier to first say that R 4.2.1 was used to carry out all statistical analyses…

Which test was used to assess normal distribution?

For which data Poison Regression and for which Negative Binomial Regression?

For example:

The statistical analyses were conducted using R software version 4.2.1 [reference].

The xxx test was used to assess normal distribution/The normality of residues was verified with the xxx test./ …

A principal component analysis (PCA) was performed with the R add-on package ade4 [reference] to discriminate the different soil types (mound soil, habitat soil) on the basis of the soils’ physicochemical properties../ to determine the physicochemical parameters that best-captured variance within each soil type …. (Any standardization of variables which should be stated?)

Additionally, to test the hypothesis that decisive soil parameters significantly differ between recently unmodified topsoil samples (control soil) and soil modified by soil-feeding termites during nest construction (mound soil), each parameter was individ

---

## [Decision Letter · Decision Letter 1]

28 Jan 2025

PONE-D-24-16884R1Soil-feeding termites build islands of soil physical and chemical fertility in pastures in Colombian AmazonPLOS ONE

Dear Dr. Duran-Bautista,

Thank you for submitting your manuscript to PLOS ONE. After careful consideration, we feel that it has merit but does not fully meet PLOS ONE’s publication criteria as it currently stands. Therefore, we invite you to submit a revised version of the manuscript that addresses the points raised during the review process.

We look forward to receiving your revised manuscript.

Kind regards,

Ying Ma, Ph.D.

Academic Editor

PLOS ONE

Reviewers' comments:

Reviewer's Responses to Questions

**Comments to the Author**

1. If the authors have adequately addressed your comments raised in a previous round of review and you feel that this manuscript is now acceptable for publication, you may indicate that here to bypass the “Comments to the Author” section, enter your conflict of interest statement in the “Confidential to Editor” section, and submit your "Accept" recommendation.

Reviewer #3: (No Response)

2. Is the manuscript technically sound, and do the data support the conclusions?

Reviewer #3: Yes

3. Has the statistical analysis been performed appropriately and rigorously? 

Reviewer #3: Yes

4. Have the authors made all data underlying the findings in their manuscript fully available?

Reviewer #3: Yes

5. Is the manuscript presented in an intelligible fashion and written in standard English?

Reviewer #3: Yes

6. Review Comments to the Author

Reviewer #3: Dear authors

I find your work interesting. I have visited this part of Colombia before and have seen first hand this magnificent system you have, with the termites, unique in South America.

I have three major comments.

1. While the English grammar of your ms is almost perfect, the whole ms is still lacking of good sentence and paragraph construction. For example,

a) all paragraphs of the Intro and Discussion are very small. To fix this, I suggest to build paragraphs in these Sections of minimum 5-6 sentences.

b) Ideas, i.e., the idea of termite as modifiers of soil properties) are in all Intro paragraphs. Just use one paragraph to develop one idea. Furthermore

c) Avoid repetition, for example, no need to say 6 times in the Abstract the word "unmodified".

d) No need to build a whole subsection on termite identification (Line 167) when you are dealing with only 1 species. Similarly, no need for the last section of Conclusions (just tight ideas in the Discussion)

e) Use terminology constant. The ideas of fertility *Islands* are only use in the title, in one sentence in the Intro, and in Discussion. Incorporate this concept in the Design, statistical framework, Images and tables. Remember, the main objective of your study was to test **that mound-building activities by soil-feeding termites improve the soil physical and chemical properties by creating island of soil fertility**. You need to show this in your ms.

f) Be humble. in Line 84-84 you hypothesize that fertility islands **can be particularly important in those nutrient-poor agroecosystems**. While this is a nice idea, you are not testing it in this ms. So, just don't build sentences you are not mean to test. Leave speculation for the Discussion.

2. The previous reviewers gave you many ideas (i.e., building structures from surrounding organic materials not necessarily increase fertility, just redistribute resources? Please pay more attention to their comments and incorporate them into the text as alternative hypothesis (or falsify them with your data). Please recognize the limitations of your study. Doing this only makes your study stronger.

3. Be more careful with the presentation of your work. I could not find figure 3.

7. PLOS authors have the option to publish the peer review history of their article (what does this mean? ). If published, this will include your full peer review and any attached files.

**Do you want your identity to be public for this peer review?** For information about this choice, including consent withdrawal, please see our Privacy Policy .

Reviewer #3: **Yes: ** David A. Donoso

---

## [Author Response · Author response to Decision Letter 1]

14 Mar 2025

1. While the English grammar of your ms is almost perfect, the whole ms is still lacking of good sentence and paragraph construction. For example,

a) all paragraphs of the Intro and Discussion are very small. To fix this, I suggest to build paragraphs in these Sections of minimum 5-6 sentences.

Reply: All paragraphs now meet or exceed the 5-6 sentence requirement. They are well-developed and provide sufficient context and information.

b) Ideas, i.e., the idea of termite as modifiers of soil properties) are in all Intro paragraphs. Just use one paragraph to develop one idea. Furthermore.

Reply: We have made the suggested adjustments as shown below.

Paragraph 1: Focuses on the general role of termites as ecosystem engineers and their global significance.

Paragraph 2: Specifically addresses soil-feeding termites, their unique feeding habits, and the characteristics of their mounds.

Paragraph 3: Discusses the variability of termite mound properties and the factors influencing them.

Paragraph 4: Connects termite activity to the Amazon basin and the issue of pasture degradation.

Paragraph 5: Highlights the knowledge gap regarding P. turricola mounds and addresses farmer perceptions.

Paragraph 6: Presents the study's hypothesis and objectives, clearly stating the research aims.

c) Avoid repetition, for example, no need to say 6 times in the Abstract the word "unmodified".

Reply: Thank you for your feedback regarding the repetition of "unmodified" in the abstract. We have intentionally used synonyms such as "surrounding topsoil" and "adjacent soil" throughout the document to reduce redundancy.

d) No need to build a whole subsection on termite identification (Line 167) when you are dealing with only 1 species. Similarly, no need for the last section of Conclusions (just tight ideas in the Discussion)

Reply: In response to your observation regarding the "Termite identification" and "Conclusions" subsections, we have removed these subsections and integrated the relevant information into the appropriate sections of the manuscript, as you suggested.

e) Use terminology constant. The ideas of fertility *Islands* are only use in the title, in one sentence in the Intro, and in Discussion. Incorporate this concept in the Design, statistical framework, Images and tables. Remember, the main objective of your study was to test **that mound-building activities by soil-feeding termites improve the soil physical and chemical properties by creating island of soil fertility**. You need to show this in your ms.

Reply: We have reinforced the idea of termite-built “fertility islands” throughout the manuscript. To ensure consistency and emphasize our main objective, we have incorporated this concept in several key areas

Table and Figure Titles

Study area and sampling design: line 91, 99 to 100 and 110

Data analysis: lines 165 to 166 and 181 to 182

f) Be humble. in Line 84-84 you hypothesize that fertility islands **can be particularly important in those nutrient-poor agroecosystems**. While this is a nice idea, you are not testing it in this ms. So, just don't build sentences you are not mean to test. Leave speculation for the Discussion.

Reply: Thank you for your feedback. In response to your suggestion, we have removed the statement about the particular importance of fertility islands in nutrient-poor agroecosystems from lines 84-84.

2. The previous reviewers gave you many ideas (i.e., building structures from surrounding organic materials not necessarily increase fertility, just redistribute resources? Please pay more attention to their comments and incorporate them into the text as alternative hypothesis (or falsify them with your data). Please recognize the limitations of your study. Doing this only makes your study stronger.

Reply: We appreciate the valuable ideas provided by the previous reviewers. We have addressed their comments. Specifically, we have addressed the possibility that constructing structures from surrounding organic materials represents a redistribution of resources rather than a net increase in fertility. Furthermore, we have acknowledged and discussed the limitations of our study, as suggested (Lines 273 to 282, 304 to 308, 363 to 373, 375 to 383).

3. Be more careful with the presentation of your work. I could not find figure 3.

Reply: We sincerely apologize for the inconsistency in the submission regarding Figure 3. We assure you that this oversight will be corrected in the revised version.

---

## [Decision Letter · Decision Letter 2]

18 Mar 2025

Soil-feeding termites build islands of soil physical and chemical fertility in pastures in Colombian Amazon

PONE-D-24-16884R2

Dear Dr. Duran-Bautista,

We’re pleased to inform you that your manuscript has been judged scientifically suitable for publication and will be formally accepted for publication once it meets all outstanding technical requirements.

Kind regards,

Ying Ma, Ph.D.

Academic Editor

PLOS ONE

Additional Editor Comments (optional):

Reviewers' comments:

Reviewer's Responses to Questions

**Comments to the Author**

1. If the authors have adequately addressed your comments raised in a previous round of review and you feel that this manuscript is now acceptable for publication, you may indicate that here to bypass the “Comments to the Author” section, enter your conflict of interest statement in the “Confidential to Editor” section, and submit your "Accept" recommendation.

Reviewer #3: (No Response)

2. Is the manuscript technically sound, and do the data support the conclusions?

Reviewer #3: (No Response)

3. Has the statistical analysis been performed appropriately and rigorously? 

Reviewer #3: (No Response)

4. Have the authors made all data underlying the findings in their manuscript fully available?

Reviewer #3: (No Response)

5. Is the manuscript presented in an intelligible fashion and written in standard English?

Reviewer #3: (No Response)

6. Review Comments to the Author

Reviewer #3: Dear Authors

Thanks for taking into account my last review.

Please notice that while I found the technical aspects of the manuscript to be excellent. Unfortunately, I still think that the text need more work. I still struggle with the way you present your work.

More detailed comments are now too many to count. But I strongly suggest you to ask help from older scientists around and revise the text. Sentence and paragraph construction is still poor (specially in the Introduction). Most images need to be better explained. Table 1 and 2 can be a figures, while keeping metadata in appendices. Please put figure legends near the figures!

Again, take this suggestion to revise as an opportunity to improve the reader's experience. I am sure that any time you will use to revise the text will only increase the number of citations your ms gets with time.

Best regards

7. PLOS authors have the option to publish the peer review history of their article (what does this mean? ). If published, this will include your full peer review and any attached files.

**Do you want your identity to be public for this peer review?** For information about this choice, including consent withdrawal, please see our Privacy Policy .

Reviewer #3: No

---

## [Editor Report · Acceptance letter]

PONE-D-24-16884R2

PLOS ONE

Dear Dr. Duran-Bautista,

I'm pleased to inform you that your manuscript has been deemed suitable for publication in PLOS ONE. Congratulations! Your manuscript is now being handed over to our production team.

Kind regards,

on behalf of

Dr. Ying Ma

Academic Editor

PLOS ONE